# Explainable Thyroid Cancer Diagnosis Through Two-Level Machine Learning Optimization with an Improved Naked Mole-Rat Algorithm

**DOI:** 10.3390/cancers16244128

**Published:** 2024-12-10

**Authors:** Wojciech Książek

**Affiliations:** Department of Computer Science, Faculty of Computer Science and Telecommunications, Cracow University of Technology, Warszawska 24, 31-155 Cracow, Poland; wojciech.ksiazek@pk.edu.pl

**Keywords:** naked mole-rat algorithm, thyroid cancer, parameter optimization, LightGBM, thyroid disease

## Abstract

This study aimed to develop new machine learning models to support thyroid cancer diagnosis by assessing tumor malignancy. The research utilized a publicly available dataset containing patient data from Shengjing Hospital of China Medical University. The primary innovation of this study lies in applying the naked mole-rat algorithm, a bio-inspired metaheuristic method, for classifier parameter optimization and feature selection. This approach led to the development of an enhanced version of the LightGBM algorithm, achieving a classification accuracy of 81.82% and an F1-score of 86.62%. Additionally, explainability analysis of the model was conducted using SHAP values.

## 1. Introduction

Cancer remains one of the leading causes of death worldwide. This article focuses specifically on thyroid cancer, the most common malignant tumor of the endocrine system, which affects the thyroid gland. As with most cancers, early detection enables effective treatment options that can significantly extend life and even offer the possibility of full recovery [1]. Over the past two decades, the incidence of thyroid cancer has risen significantly. For instance, ref. [2] reported a 3% increase in thyroid cancer incidence in the United States from 1974 to 2013, while [3] highlighted it as the fastest-growing cancer in the UK, with an average annual percentage change of 2.49% in men and 2.34% in women. In 2020, thyroid cancer accounted for 570,000 cases worldwide, making it the 10th most common cancer globally [4], with 77% of cases occurring in women. In Europe, 78,000 cases and 7000 deaths were reported [1]. Table 1 presents detailed statistics on thyroid cancer incidence and mortality for 2022.

It can be observed that Asia had the highest number of thyroid cancer cases and deaths. Furthermore, the total number of cases increased by more than 200,000 between 2020 and 2022, representing a significant rise. Data from 2022 [5] indicate that thyroid cancer was the third most common cancer in China and Saudi Arabia, the fourth in Mexico, and the fifth in Brazil and Turkey. Comprehensive incidence data by country are provided in the Appendix A. Projections by ref. [4] estimate that by 2040, thyroid cancer cases will reach 761,000, with an estimated 72,900 deaths. However, the number of detected cases in 2022 had already surpassed these predictions. According to the latest research [6], 43,720 cases of thyroid cancer were diagnosed in the USA in 2023, representing a 313% increase in incidence over recent decades. The American Cancer Society predicts a slight increase in 2024, with the number of detected cases expected to reach 44,020 [7]. Factors contributing to thyroid cancer include exposure to ionizing radiation, such as from radiotherapy or nuclear fallout. A family history of the disease also increases the risk, and genetic conditions such as Cowden’s disease and multiple endocrine neoplasia type 2 (MEN2) are recognized risk factors [4]. Studies indicate that cancer risk increases significantly in patients who undergo radiotherapy or chemotherapy before the age of 35 [8], particularly in those with a prior history of breast cancer [9].

This research aimed to contribute to the progress in both medical diagnostics and bioinspired metaheuristics by addressing advanced optimization problems, such as parameter tuning for machine learning models and feature selection. The main objectives of this research were as follows:Developing an effective classification model to support thyroid cancer diagnosis by optimizing parameters and selecting features.Adapting and applying the naked mole-rat algorithm for parameter optimization and feature selection of classification models, comparing its performance to that of the default classifier settings.Focusing on model explainability by utilizing the SHAP values method.

## 2. Literature Review

The ongoing rise in cancer cases and the resulting increase in mortality create a significant burden on healthcare systems, exposing shortages in both equipment and skilled personnel necessary for patient treatment. One potential solution to alleviate this burden and support clinicians and diagnosticians is the application of advanced technologies such as artificial intelligence (AI), particularly machine learning, in medical diagnostics and therapy planning [10]. For example, in [11], a new machine learning model based on nested ensemble learning was developed to predict breast cancer using the Wisconsin Diagnostic Breast Cancer (WDBC) dataset, achieving a high classification accuracy of 98.07%. In [12], a machine learning model incorporating two-level feature selection via neighborhood component analysis (NCA) and a genetic algorithm (GA) was designed. The Nu-SVM algorithm was selected for classification, and the model achieved a classification accuracy of 96.36% in predicting the survival of hepatocellular carcinoma patients using data from Coimbra’s Hospital and University Center in Portugal. In [13], a new lightweight convolutional neural network model for detecting brain tumors was proposed, achieving 99.48% accuracy for binary classification and 96.86% for multi-class classification. This demonstrates that simpler models can achieve high accuracy, without requiring extensive computational resources. Machine learning models have also been effective in predicting survival outcomes for colon cancer patients, achieving approximately 80% accuracy for one-year, three-year, and five-year survival predictions. These models utilized gradient boosting classifiers optimized with advanced techniques like Optuna, HyperOpt, or Raytune [14]. AI methods have also been applied specifically to thyroid cancer research. In [15], machine learning techniques were employed to predict distant metastasis of thyroid cancer using data from the NIH SEER database, where the best model—a random forest—achieved a classification accuracy of 90.6% and an F1-score of 90.8%. In [16], convolutional neural networks were used to diagnose thyroid cancer from sonographic images, yielding a classification accuracy of 89% on a large dataset with over 130,000 images of affected individuals and over 180,000 images of healthy individuals. Further research on thyroid cancer includes [17], where MRI images were analyzed for early detection of the disease using convolutional networks, achieving an accuracy of 87%. In [18], a mask R-CNN convolutional network was developed to detect nodules in ultrasound images, achieving 84% precision and 79% recall on a dataset with 821 images. In [19], deep learning, bio-inspired metaheuristics, and MCDM algorithms were combined to detect thyroid abnormalities from ultrasound and histopathological images, achieving a classification accuracy of 99.13% using the Fox algorithm, PCA, and a random forest model. In [20], machine learning models were used to predict recurrence risk in patients with well-differentiated thyroid cancer, with the support vector machine (SVM) achieving the highest performance: sensitivity of 99.33%, specificity of 97.14%, and AUC of 99.71%. The study by [21] employed multi-channel convolutional networks, specifically the Xception network, to diagnose thyroid cancer. The research was conducted using both ultrasound and computed tomography (CT) images. The dataset included 917 ultrasound images and 2352 CT images. The classification accuracy achieved was 98.9% for ultrasound images and 97.5% for CT images. In [22], the authors conducted their research on a dataset comprising histopathological images, which included 11,715 images from 806 patients. For classification, they utilized deep learning methods, specifically the Inception-ResNet-v2 and VGG-19 convolutional networks. The study focused on a multi-class classification task, addressing the following classes: normal tissue, adenoma, nodular goiter, papillary thyroid carcinoma (PTC), follicular thyroid carcinoma (FTC), medullary thyroid carcinoma (MTC), and anaplastic thyroid carcinoma (ATC). The VGG-19 model achieved a classification accuracy of 97.34%, while Inception-ResNet-v2 achieved 94.42%. These results demonstrate that deep learning models can be effectively applied to complex diagnostic challenges in thyroid cancer.The researchers in [23] conducted an intriguing study aimed at predicting the BRAFV600E mutation in thyroid cancer using ultrasound images. The study utilized a dataset comprising 14,194 ultrasound images and employed several pre-trained deep learning models, including AlexNet, GoogLeNet, SqueezeNet, and Inception-ResNet-v2. The best-performing model achieved a classification accuracy of 57.7% and an AUC of 64.6%. Another study [24] focused on distinguishing between malignant and benign thyroid tumors using data from 1232 patients and various classifiers, with random forest achieving the best results: classification accuracy of 78.01% and AUC of 84.43%. Current methods for assessing nodule malignancy heavily rely on the clinical experience of radiologists, leading to potential diagnostic errors. This is a key area where machine learning can enhance diagnostic accuracy.

A summary of selected literature on the application of machine learning methods to thyroid-cancer-related issues is presented in Table 2. It is important to note that these studies were conducted using various types of data (e.g., tabular data, ultrasound, MRI, CT, histopathological images) and pursued different objectives, such as disease diagnosis, prediction of metastases, assessment of recurrence risk, and evaluation of tumor malignancy. Consequently, direct comparison of the results is not feasible. However, it is worth highlighting that the number of studies in this area remains relatively small. Further research is essential to develop effective machine learning models for diagnosing and supporting the treatment of patients with suspected or confirmed thyroid cancer.

This article focuses on developing new machine learning models to aid in thyroid cancer diagnostics, specifically in determining whether a tumor is malignant or benign. The research was based on a dataset provided by [24]. It is important to note that, unfortunately, a significant portion of published scientific papers do not make their experimental data publicly available, which naturally hinders the continuation and replication of certain studies. The primary motivation for this research was the necessity of developing new machine learning models capable of effectively assessing the malignancy of thyroid tumors. The study utilized the dataset from [24] to ensure that future researchers can compare their work, facilitating the development of more effective diagnostic models. It is worth emphasizing that the number of scientific studies focusing on this issue is relatively small, underscoring the need for continued efforts in this field to design models that could eventually be applied in clinical practice. Furthermore, previous studies have often overlooked the aspect of model explainability, which is essential for the application of machine learning in medical diagnostics. An additional motivation for the author was to advance modern bio-inspired metaheuristic methods, such as the naked mole-rat algorithm, to demonstrate their effectiveness and highlight their value in addressing machine learning challenges in medical diagnostics.

## 3. Materials and Methods

### 3.1. Dataset

The author utilized a dataset published as part of a 2022 study conducted in China [24]. The data were collected at Shengjing Hospital of China Medical University between 2010 and 2012. The dataset comprises 1232 records, each containing 19 features, including demographic information, ultrasound characteristics, and various test results. The classification problem addressed in this study is binary, distinguishing between malignant and benign cases. The dataset comprises 819 malignant samples and 413 benign samples. A detailed description of these features is provided in Table 3.

Figure 1 shows the gender distribution of patients who participated in this study.

Analysis of Figure 1 reveals that a substantial majority of patients were women, comprising 83.77% of the total, while men accounted for the remaining 16.23%. Figure 2 illustrates the age distribution of these patients.

Analysis of Figure 2 shows that the average patient age was 46.61, with an age range of 13 to 82 years. Further details about the dataset are available in the work by Xi et al. [24].

### 3.2. Machine Learning Algorithms

Ten widely used classification algorithms, renowned for their high effectiveness in medical applications, were selected for the experiment. These methods included decision tree [26], extra tree [27], gradient boosting [28], nearest neighbor algorithm [29], LightGBM [30], logistic regression [31], NU-SVM support vector machine [32], random forest [33], XGBoost [34], and extra trees [35] models. Deep learning methods were not used in these studies due to the small sample size and the limited number of features in the dataset. Each classifier has a set of parameters that should be optimally tuned to maximize the classification efficiency, as using default parameters generally does not yield the highest performance. Table 4 presents the optimized parameters for each classifier selected. The Appendix A also include the value ranges for the individual parameters used in the optimization process.

### 3.3. Parameter Optimization and Feature Selection

Optimizing machine learning model parameters is a crucial step in developing effective models. Typically, standard algorithms such as grid search or random search are used. In this study, the author proposes framing parameter tuning and feature selection as an optimization problem and leveraging biologically inspired metaheuristic algorithms to solve it. These algorithms generally operate by generating an initial population of individuals (each representing a potential solution), which is then iteratively refined to approach the optimal solution [36]. Despite their non-deterministic nature, these algorithms have gained popularity due to their strong performance across various fields, including healthcare, engineering, mathematics, economics, linguistics, and science [37]. Among the most popular of these algorithms are the genetic algorithms, which are based Darwinian evolution principles, generating new individuals through crossover and mutation operators [38]. Genetic algorithms have proven effective for machine learning parameter tuning and feature selection. For instance, they were used to develop a classification model for myocardial dysfunction diagnosis, achieving an accuracy of 91.40% [39]. A similar approach achieved high classification accuracy (88.24%) in diagnosing schizophrenia based on EEG signals [40]. Genetic algorithms have also contributed to cancer diagnosis research, with one model that used genetic algorithms and chemical reaction optimization achieving 99.64% accuracy on the WBC dataset [41]. These studies have frequently employed widely used classifiers, such as XGBoost and support vector machines. The author also has experience in effectively applying genetic algorithms for developing machine learning models for tasks such as hyperspectral data analysis [42], predicting survival in hepatocellular carcinoma patients [43], and diagnosing coronary artery disease [44].

For this study, the author selected an alternative to genetic algorithms: the naked mole-rat (NMR) algorithm [45]. This decision was based on several factors. First, the NMR algorithm is relatively new and has not been extensively explored for parameter tuning and feature selection. Second, it has demonstrated superior performance in benchmark tests, surpassing established algorithms such as particle swarm optimization (PSO), grey wolf optimization (GWO), whale optimization (WOA), differential evolution (DE), gravitational search (GSA), fast evolutionary programming (FEP), the bat algorithm (BA), flower pollination algorithm (FPA), and firefly algorithm (FA). Inspired by these promising results, the author conducted this study using the NMR algorithm, with minor enhancements, including mutation probability, a crossover operator, and the Levy flight technique. This adapted algorithm was tailored for parameter optimization and feature selection, as shown in Figure 3.

Figure 3 illustrates an example of an individual for the KNN algorithm, showcasing two approaches to the problem: parameter optimization alone and parameter optimization combined with feature selection. In this algorithm, an individual is represented as a list of real numbers, which are then decoded into the relevant parameters for the classification model. For instance, the KNN algorithm can be configured with different metrics, such as Euclidean, Manhattan, and Chebyshev. Encoding this parameter is straightforward: values in the range [0, 1) correspond to Euclidean, [1, 2) to Manhattan, and [2, 3) to Chebyshev. Similarly, the weights parameter is encoded as follows: values in the range [0, 1) represent ‘uniform’, and [1, 2) represent ‘distance’. The n_neighbors parameter, which specifies the number of neighbors, is an integer. To encode it, a real number between 1 and 10 is randomly selected and then cast to an integer. For feature selection, additional elements are added to the list, with one element for each feature. If a feature’s value falls within [0, 0.5), it is excluded from the model; if it falls within [0.5, 1), it is included. This approach makes it easy to adapt the NMR algorithm for both parameter optimization and feature selection, a strategy that can be generalized across classification and regression models.

The author applied this individual structure to ten different classification models, as discussed in the previous section. For more complex algorithms, such as XGBoost or LightGBM, the individual includes additional elements, making the parameter selection process more intricate, in order to achieve optimal performance.

Configuring the NMR algorithm itself is relatively simple and requires setting the following parameters:epoch—200pop_size—1000pb (breeding probability)—0.75pm (probability of mutation)—0.1

Multithreaded processing was used during the calculations to parallelize the evaluation of the fitness function for each individual in the population, which significantly reduced the computation time.

### 3.4. Metrics

In the conducted research, standard metrics commonly used in classification problems—such as accuracy, F1-score, precision, and recall—were utilized. These metrics were calculated using the test set. The formulas for these metrics are as follows:(1)$$Accuracy=\frac{TP+TN}{TP+TN+FP+FN}$$
(2)$$Precision=\frac{TP}{TP+FP}$$
(3)$$Recall=\frac{TP}{TP+FN}$$
(4)$$F1=\frac{2*Precision*Recall}{Precision+Recall}=\frac{2*TP}{2*TP+FP+FN}$$
where TP is the number of true positives, TN is the number of true negatives, FP is the number of false positives, FN is the number of false negatives.

Additionally, this article includes ROC curves and confusion matrices for the top-performing models. For the other models, this information is available in the Appendix A.

### 3.5. Experiment Schema

To ensure that the journal’s readers fully understand the experiment, the author has prepared a graphical diagram illustrating the key elements of the research conducted.

Figure 4 illustrates the experimental design. The research utilized a dataset provided by Shengjing Hospital of China Medical University, consisting of 1232 records. The ID column was removed, and the data were standardized. Consistent with the original study that presented and made this dataset available, the author employed 10-fold cross-validation to divide the data into training and test sets. The use of 10-fold cross-validation not only facilitated the comparison of the obtained results with those of [24] but also helped reduce overfitting in the [46] models. However, the author highlights in the study’s limitations that further development of these models will require access to larger datasets. Ten well-known and effective classification algorithms were utilized in the research: decision tree, extra tree, gradient boosting, K-nearest neighbors (KNN), LightGBM, logistic regression, NuSVM, random forest, XGBoost, and extra trees.

The primary novelty of this study lies in the application of the naked mole-rat algorithm in two approaches:1-level optimizer: Optimization of Classifier Parameters.2-level optimizer: Optimization of Classifier Parameters and Feature Selection.

An additional experiment was conducted using the default classifier parameters and the full set of features to compare the final results. The outcomes were primarily measured using the classification accuracy and F1-score calculated on the test set. For the top-performing models, confusion matrices and ROC curves were generated, and an explainability analysis was performed using SHAP values. Additionally, the SMOTE [47] and ADASYN [48] techniques were applied to balance the dataset for the best models obtained through two-level parameter optimization and feature selection.

## 4. Results

This section presents the results of the conducted research, organized into four subsections: classifiers with default parameters, a 1-level optimizer for classifier parameter optimization, a 2-level optimizer for parameter optimization and feature selection, and model explainability. The research was carried out using a dataset provided by Shengjing Hospital of China Medical University. The data underwent appropriate preprocessing and rescaling before the experiments commenced, and the naked mole-rat algorithm, a metaheuristic inspired by biological processes, was then applied for optimization.

A total of ten classification algorithms were employed, and the entire study was conducted using 10-fold cross-validation. The primary metrics for model evaluation included classification accuracy, F1-score, AUC, recall, and precision. Additionally, for the best-performing models, confusion matrices and ROC curves were generated. The corresponding materials for the other models can be found in the Appendix A. The experiments were conducted on hardware with the following specifications:Processor: Intel(R) Xeon(R) Gold 6258R CPU @ 2.70 GHz 2.69 GHz (2 processors)Installed RAM: 1.50 TBWindows Specifications: Windows Server 2022 Standard

The source code for the experiments was prepared in Python using additional packages such as: Numpy [49], Pandas [50], Mealpy [51], Scikit-Learn [52], Imbalanced-learn [53].

### 4.1. Classifiers with Default Parameter Values

This section presents the results for classifiers using their default parameter values. The experiments were conducted with the following classifiers: decision tree, extra tree, gradient boosting, K-nearest neighbors (KNN), LightGBM, logistic regression, NuSVM, random forest, XGBoost, and extra trees.

Table 5 presents the results obtained for classifiers using the default parameters. In the first experiment, the author chose not to apply parameter optimization, in order to observe the performance of the classifiers with their default settings, as configured in libraries such as scikit-learn, XGBoost, and LightGBM. In subsequent sections of the results, the improvements in model performance achieved through parameter optimization using the naked mole-rat algorithm will be highlighted. The classifier with the highest classification accuracy was the logistic regression model, achieving an accuracy of 0.7792 and an F1-score of 0.8354. Closely following was the NuSVM support vector machine, with a classification accuracy of 0.7752 and an F1-score of 0.8313. The next highest-performing classifier was gradient boosting, which achieved an accuracy of 0.7735 and an F1-score of 0.8331. In contrast, the decision tree classifier exhibited the lowest performance among all the tested models, with a classification accuracy of 0.6988.

Figure 5 displays the confusion matrix for the best-performing model in this section, which was logistic regression. This model demonstrated strong performance in classifying both classes, achieving a final classification accuracy of 0.7792. Figure 6 presents the ROC curve for this model, with an AUC value of 0.84.

### 4.2. 1-Level Optimizer—Optimization
of Classifier Parameters

This section presents the results obtained using the naked mole-rat algorithm to optimize classifier parameters. In contrast to the experiments in the previous section, these experiments involved optimizing the parameters of the machine learning models to enhance their classification performance. For each classifier, key parameters for optimization were selected and encoded as real individuals within the population of this biology-inspired algorithm. The population size was set to 1000, and the algorithm was run for 200 epochs. This approach resulted in improved performance across all classification models used in the study.

Table 6 presents the results for the classifiers optimized using the naked mole-rat algorithm. A comparison with the results obtained using default parameter values reveals a significant improvement. The XgBoost algorithm achieved the best performance, with a classification accuracy of 0.8077 and an F1-score of 0.8595. In contrast, the accuracy of this model with default parameters was only 0.7451, indicating that the appropriate selection of parameters enhanced its accuracy by over 6%. Classifiers within the gradient boosting family are typically known for their high efficiency on tabular data; however, they require careful parameter selection and substantial computing power, as demonstrated by this research. Other classifiers that achieved a classification accuracy above 80% included extra trees, random forest, gradient boosting, and LightGBM.

Figure 7 presents the confusion matrix for the best model obtained through parameter optimization using the naked mole-rat algorithm. The XgBoost algorithm emerged as the top performer, and the confusion matrix highlights its strong classification performance for both recognized classes. Figure 8 shows the ROC curve for this model, which achieved a high AUC value of 0.85. This performance indicates that the model is well-suited for clinical applications.

### 4.3. 2-Level Optimizer—Optimization
of Classifier Parameters and Feature Selection

In the third phase of the experiments, the naked mole-rat algorithm was enhanced to not only optimize parameters but also select the optimal set of features to maximize the classification accuracy. This modification involved extending the individual representation in the algorithm to include an 18-element feature vector. Each feature value ranged [0, 1], where values in the range [0, 0.5) indicated that the feature would be excluded from the model, while values in the range [0.5, 1] indicated that the feature would be included. This approach resulted in an improvement in classification accuracy across all models used in the study.

Table 7 presents the results obtained from the two-level optimizer, which focused on both classifier parameter optimization and feature selection. This dual approach to developing machine learning algorithms for supporting thyroid cancer diagnosis resulted in a model with a classification accuracy of 0.8182 and an F1-score of 0.8662. These results were achieved using the LightGBM algorithm with 13 features from the dataset. Two other algorithms, XgBoost and random forest, also achieved classification accuracies exceeding 81%. While the improvement in results compared to parameter optimization alone was modest, it is important to note that using a smaller number of features while maintaining similar or improved effectiveness significantly enhances a model’s efficiency for training and clinical application.

Figure 9 displays the confusion matrix for the best-performing model across all conducted experiments: the LightGBM classifier optimized using the naked mole-rat algorithm in conjunction with feature selection. This model achieved a very high classification accuracy of 0.8182, demonstrating excellent performance in distinguishing between the two categories. Figure 10 presents the ROC curve for this model, which achieved the highest AUC value of 0.86 among all the studies. With such strong classification performance, this model can be effectively utilized in clinical practice to assist oncologists in assessing the malignancy of cancerous tumors.

### 4.4. Explainability of the Model

When designing machine learning models, it is crucial to consider their explainability and interpretability. This is especially important for models used in medical applications, as it allows doctors and diagnosticians to understand the reasoning behind a tool’s decisions. Historically, machine learning models were often regarded as black boxes, making their inner workings opaque. However, recent years have seen significant advancements in model explainability techniques, which have been successfully implemented in various fields. In this article, the author chose to utilize the Shap (Shapley additive explanations) values method, which is one of the most popular and widely adopted techniques in the scientific literature for explaining machine learning models used in medical contexts. Shap values have been used to clarify models related to nasopharyngeal cancer survival [54], predict the survival of patients with coronary artery disease [55], and assess the risk of breast cancer [56].

Figure 11 illustrates the impact of individual features on the model’s final results. The vertical axis lists the features included in the model, with calcification, multilaterality, and shape identified as the most significant factors influencing the model’s performance. In contrast, gender, echo strength, and FT3 were found to have the least impact. This analysis was performed for the best-performing model in the study—the LightGBM classifier, which was optimized using a two-level approach combining parameter optimization and feature selection. This model achieved a classification accuracy of 81.82.

### 4.5. Application of SMOTE and ADASYN for Data Balancing

In the final part of the experiment, the best models obtained through parameter optimization and feature selection using the naked mole-rat algorithm were applied, and the most popular oversampling techniques—SMOTE and ADASYN—were used. The goal of applying these methods was to balance the training set, ensuring that the model could effectively classify both benign and malignant tumors.

In Table 8, it can be seen that the oversampling techniques used—SMOTE and ADASYN—did not result in an improvement in model classification accuracy, and even led to a slight decrease in effectiveness. In this experiment, the XGBoost algorithm performed the best, achieving a classification accuracy of 0.7833 and an F1-score of 0.8313. The best results achieved through parameter optimization and feature selection alone were obtained by the LightGBM algorithm, with a classification accuracy of 0.8182 and an F1-score of 0.8662. Therefore, it can be concluded that the use of oversampling methods did not improve the performance of the algorithms. This may have been due to the complexity of the medical data or the relatively small dataset, which may have limited the full potential of these algorithms.

Figure 12 shows the confusion matrix for the best model from the oversampling experiment. The ADASYN model performed slightly better than the SMOTE technique, achieving a final classification accuracy of 0.7833. Additionally, Figure 13 presents the ROC curve for this model, which demonstrates a high AUC value of 0.84. Confusion matrices and ROC curves for all the models listed in Table 8 can be found in the Appendix A.

## 5. Discussion

Thyroid cancer is a group of malignancies whose incidence has been steadily increasing in recent decades, leading to a higher number of diagnoses. A significant diagnostic challenge lies in determining whether detected tumors are malignant or benign. The aim of this research was to develop machine learning models to predict the nature of these tumors. This distinction is crucial for diagnosis and subsequent treatment, as therapeutic approaches differ depending on a tumor’s characteristics. The author utilized a dataset collected at Shengjing Hospital of China Medical University, which was published alongside a scientific paper detailing the design of machine learning models based on the dataset [24]. The primary novelty of this research was the use of the naked mole-rat algorithm, a relatively new population-based algorithm within the bio-inspired metaheuristics category. To date, this algorithm has been predominantly applied to standard benchmark problems, such as the optimization of multidimensional functions in the CEC conference. Notably, bio-inspired metaheuristics have been rarely utilized in thyroid cancer detection problems—only one study using the Fox optimizer algorithm was found in the literature. This highlights the need for further research in this field, especially since these methods have demonstrated high effectiveness in numerous medical classification tasks. In this study, the naked mole-rat algorithm was applied for the first time to optimize machine learning model parameters and to perform feature selection. In the first optimization task—classifier parameter selection—a method was developed to encode parameters as real numbers, such that each individual in the population represented a classifier capable of diagnosing thyroid cancer. For certain parameters, such as the metric in the nearest neighbor algorithm (Euclidean, Manhattan, Chebyshev), text-based values were encoded as real numbers, to align with the requirements of the naked mole-rat algorithm. This approach resulted in a one-level optimization model. The second part of the research introduced a novel two-level optimization approach, which simultaneously optimized both parameters and feature selection. In this case, each individual was extended with an additional feature vector, where each feature had a value ranging from 0 to 1. Values between 0 and 0.5 indicated the rejection of a feature, while values between 0.5 and 1 signified feature selection for the model. This two-level approach allowed for the exploration of appropriate classifiers within the multidimensional space of parameters and features, presenting a complex optimization problem. It is important to note that this represents a significant advancement compared to the study in [24], where parameter selection and feature selection were not discussed. Additionally, this research involved the use of 10 distinct classification models, ensuring a comprehensive study that included both well-established algorithms based on metrics and rules, as well as effective gradient boosting models and support vector machines for small- and medium-sized datasets. Furthermore, the interpretability of the models was emphasized through the use of SHAP values, a critical aspect for implementing algorithms in clinical practice. Additional experiments were conducted with oversampling techniques, including SMOTE and ADASYN, to assess the potential of these methods for producing highly efficient classifiers. The description above highlights the high novelty of this research and the thoroughness of the experiments conducted.

Figure 14 presents a comparative summary of the results for the ten classifiers across three scenarios: default parameter values, parameter optimization, and parameter optimization with feature selection using the naked mole-rat algorithm. The results indicate that employing the naked mole-rat algorithm significantly improved the performance of the classifiers compared to using default parameters and the full feature set. The LightGBM model achieved the best results, with a classification accuracy of 0.8182, an F1-score of 0.8662, an AUC of 0.86, and a recall of 0.884. The XGBoost algorithm also delivered impressive results, achieving a classification accuracy of 0.8141, while the random forest classifier closely followed with an accuracy of 0.8133. These findings demonstrate that the naked mole-rat algorithm can be effectively applied to complex optimization tasks, such as parameter optimization and feature selection, particularly when applied simultaneously. It is important to note that these studies were made possible by the dataset provided by the authors of the work in [24]. This practice is commendable, as it allows other researchers to build upon previously proposed models and develop even more effective solutions for clinical applications, while also facilitating objective comparisons of results across studies.

Table 9 presents a comparison between the results obtained in this study and those reported in the original article, where this dataset was first introduced. Although the LightGBM algorithm was not used in that work, it achieved a classification accuracy of 81.82% in the current study. The random forest classifier also showed a notable improvement, with a classification accuracy of 81.33% compared to 78.01% in the earlier study. Similarly, the XGBoost algorithm achieved a classification accuracy of 81.41% in this research, while the original article reported a result of 77.22%. These improvements in performance can be attributed to the implementation of the naked mole-rat algorithm for parameter optimization and feature selection. Additionally, this study employed the Shap values technique to enhance the model interpretability. Furthermore, a wider range of classifiers were explored, with a total of ten different classification algorithms evaluated. Both studies employed 10-fold cross-validation, ensuring a valid comparison of the results. In this discussion, we avoid comparing our results with the studies listed in Table 9, as those studies were conducted on datasets with entirely different structures (e.g., ultrasound, MRI, CT scans, histopathological images). Models that perform well on tabular data may not necessarily achieve similar performance on image-based data. Additionally, most of the datasets referenced in these publications are not publicly available, making it impossible to evaluate newly designed models against them.

In the concluding section of this discussion, the author will highlight the key achievements of this research, as well as its limitations. The main achievements include

Adaptation of the naked mole-rat algorithm for optimizing classifier parameters (1-level optimizer).Adaptation of the naked mole-rat algorithm for optimizing classifier parameters and selecting features (2-level optimizer).Conducting experiments with ten different classification methods, including top-performing algorithms for tabular data, such as XGBoost and LightGBM.Performing an explainability analysis of the best model using Shap values.

The main limitations include

The research was conducted using a single dataset from China, comprising 1232 data samples. This dataset may be insufficient to ensure a high generalization ability for the model and could lead to a degree of overfitting. The next step in developing these models will be to create a significantly larger dataset. Furthermore, acquiring data from diverse geographical regions would be highly beneficial, as it would allow consideration of local factors such as diet, environmental conditions, and access to healthcare. Unfortunately, such data are not currently publicly available. Nevertheless, the author remains optimistic that more datasets will become accessible in the future, enabling further advancements in this research.The author focused exclusively on the naked mole-rat algorithm. In recent years, several effective biology-inspired algorithms have been developed, such as the Coati optimization algorithm [57], success history intelligent optimization [58], and fennec fox optimization [59]. These algorithms have seen limited application in conjunction with machine learning methods.The author also believes it would be worthwhile to explore the use of ensemble machine learning models based on voting or model blending (stacking). This approach may lead to even higher classification accuracy.

The research conducted demonstrated the significant potential of the naked mole-rat algorithm. Machine learning models built using this algorithm showed high classification accuracy, highlighting its potential for use in various advanced machine learning problems. These include the optimization of ensemble classifiers built using stacking methods, training neural networks (as an alternative to the standard backpropagation technique), and searching for optimal neural network architectures. Additionally, the high performance observed on the tabular dataset suggests that similar results could be achieved with more complex data, such as ECG, EEG signals, or medical imaging data like ultrasound, X-ray, or CT scans.

## 6. Conclusions

This study focused on developing effective machine learning models to aid in the diagnosis of thyroid cancer by assessing the malignancy of detected tumors. Ten classification models were built using the naked mole-rat algorithm for parameter optimization, as well as for simultaneous parameter optimization and feature selection. The most effective model achieved a classification accuracy of 81.82%. The use of the naked mole-rat algorithm for two-level optimization of machine learning model parameters and feature selection led to the achievement of high-performance results. Although the results suggest that these algorithms could be applied in clinical practice, further research is needed to improve their predictive accuracy.

## Figures and Tables

**Figure 1 cancers-16-04128-f001:**
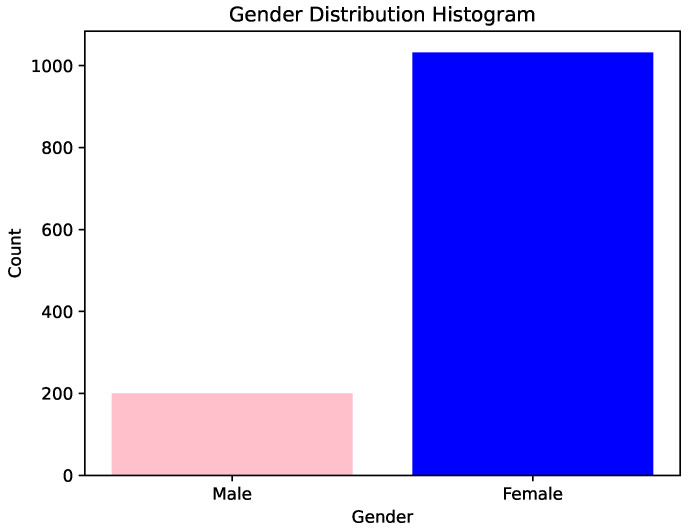
Gender distribution histogram.

**Figure 2 cancers-16-04128-f002:**
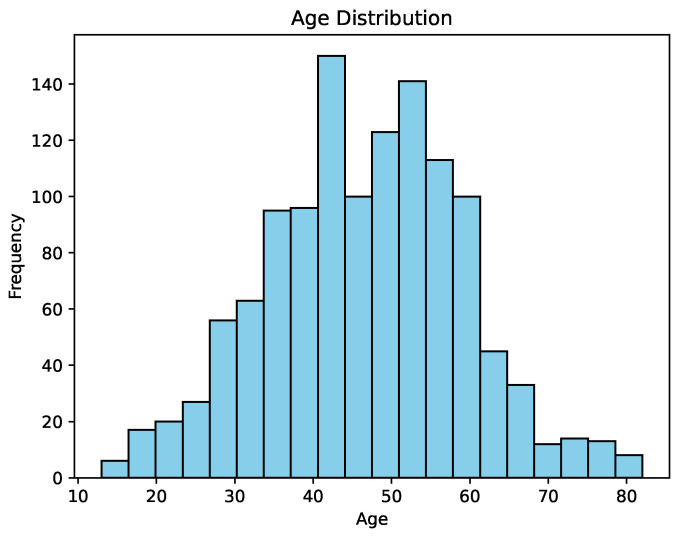
Age distribution of patients.

**Figure 3 cancers-16-04128-f003:**
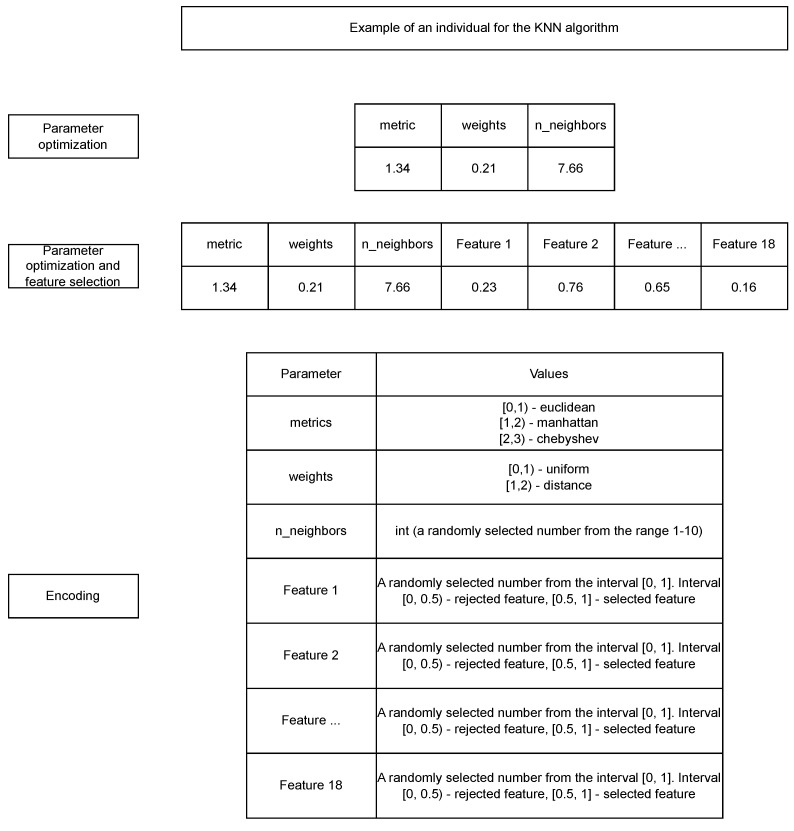
The structure of an individual in the NRM algorithm.

**Figure 4 cancers-16-04128-f004:**
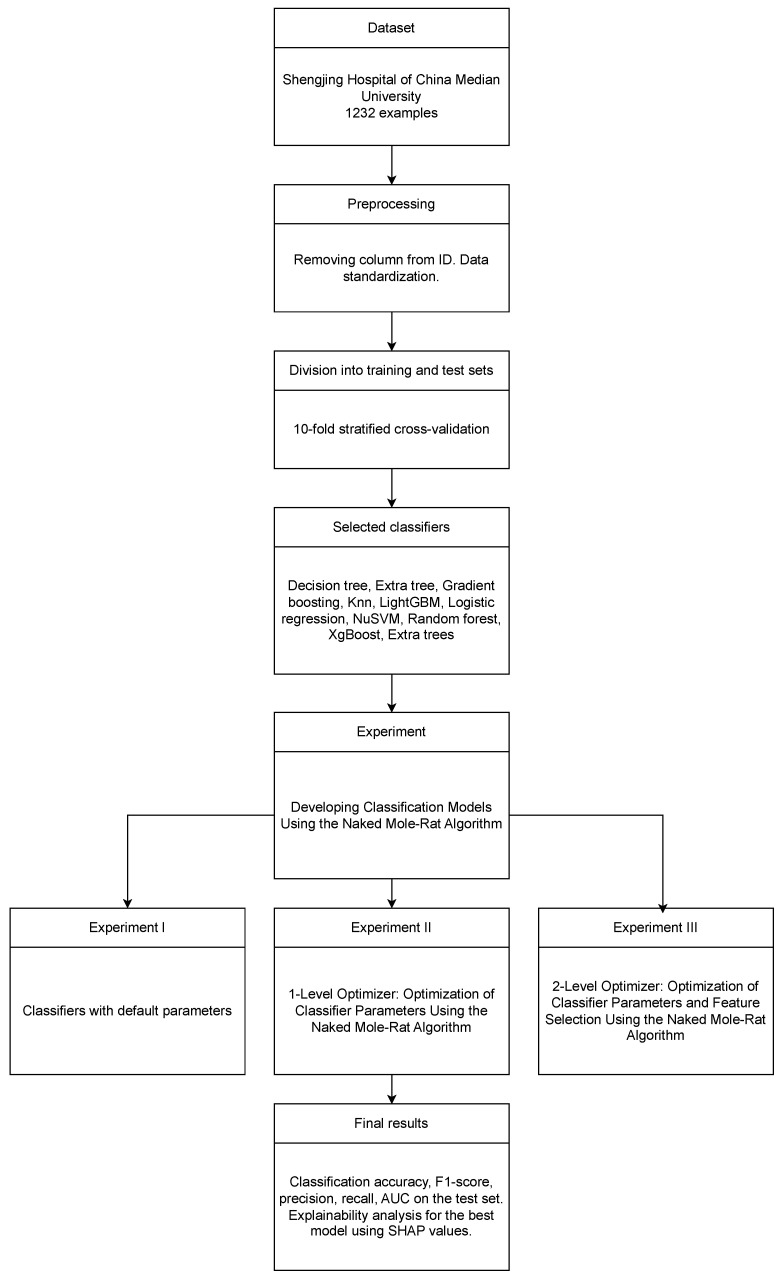
Experiment schema.

**Figure 5 cancers-16-04128-f005:**
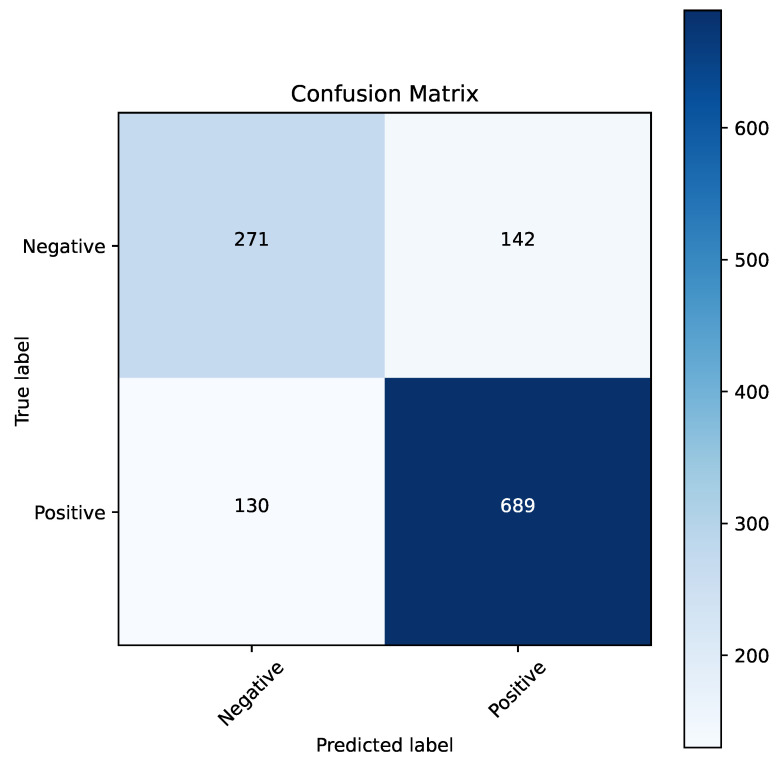
Confusion matrix for the logistic regression algorithm with default parameter values.

**Figure 6 cancers-16-04128-f006:**
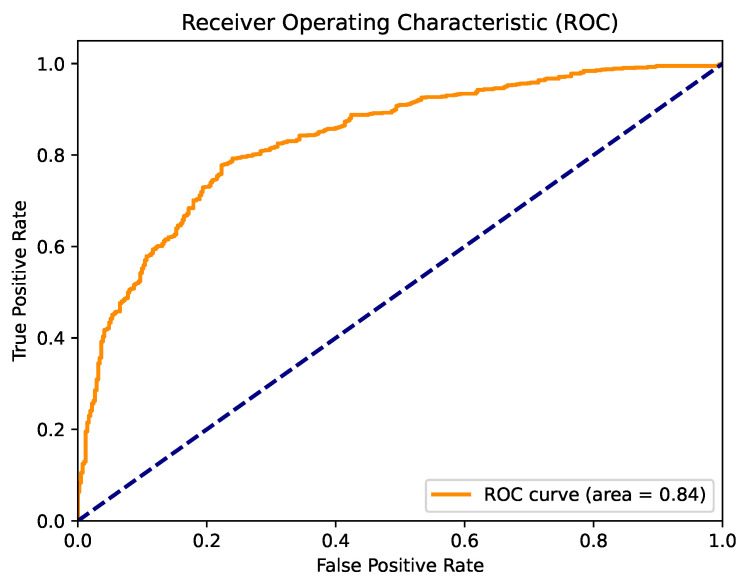
ROC curve for the logistic regression algorithm with default parameter values.

**Figure 7 cancers-16-04128-f007:**
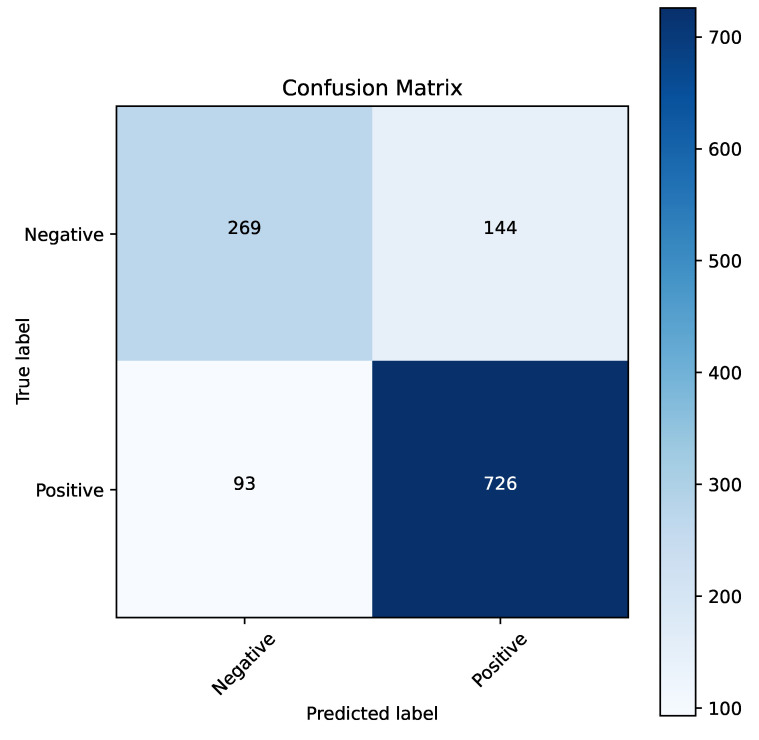
Confusion matrix for the XgBoost algorithm when optimizing parameters using the naked mole-rat algorithm.

**Figure 8 cancers-16-04128-f008:**
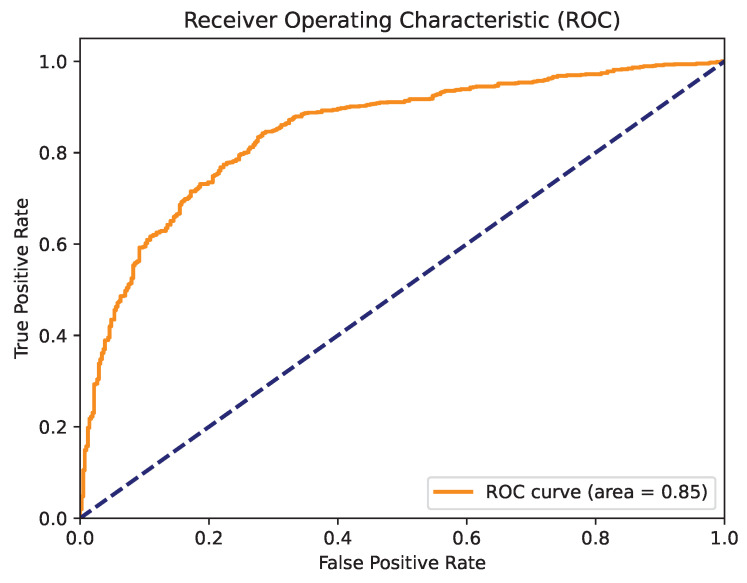
ROC curve for the XgBoost algorithm with parameter optimization using the naked mole-rat algorithm.

**Figure 9 cancers-16-04128-f009:**
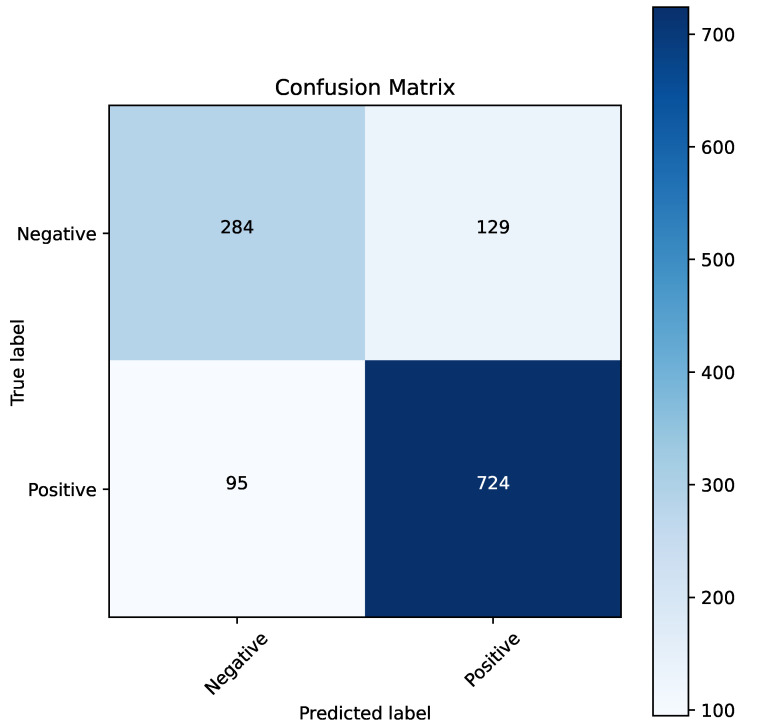
Confusion matrix for LightGBM algorithm for parameter optimization and feature selection using naked mole-rat algorithm.

**Figure 10 cancers-16-04128-f010:**
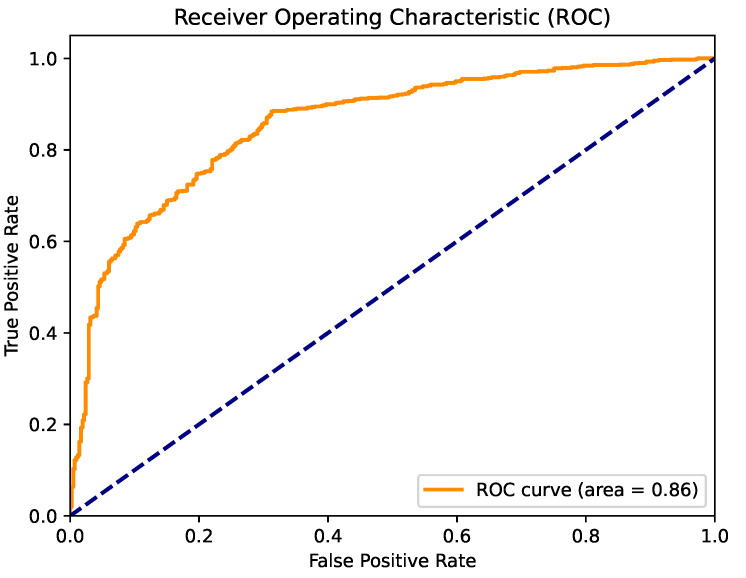
ROC curve for LightGBM algorithm with parameter optimization and feature selection using naked mole-rat algorithm.

**Figure 11 cancers-16-04128-f011:**
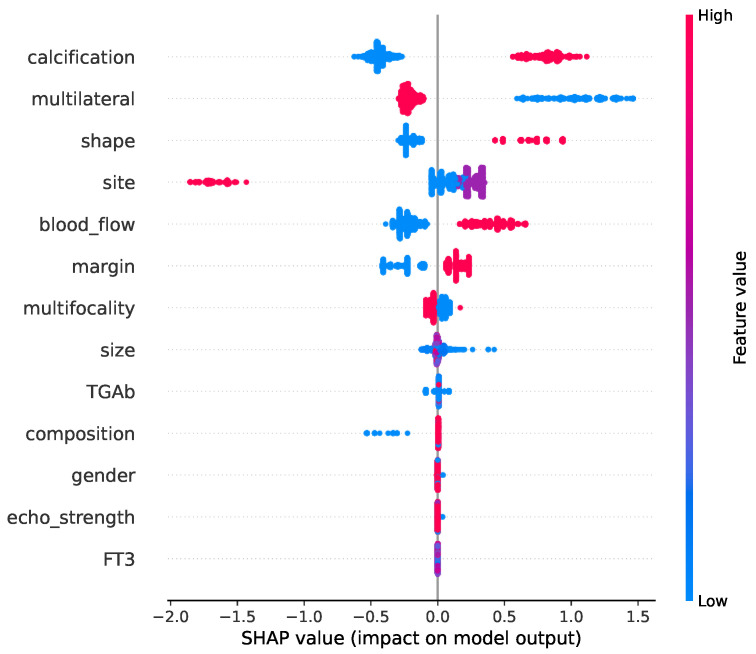
Shap values for the LightGBM model with parameter optimization and feature selection using the naked mole-rat algorithm.

**Figure 12 cancers-16-04128-f012:**
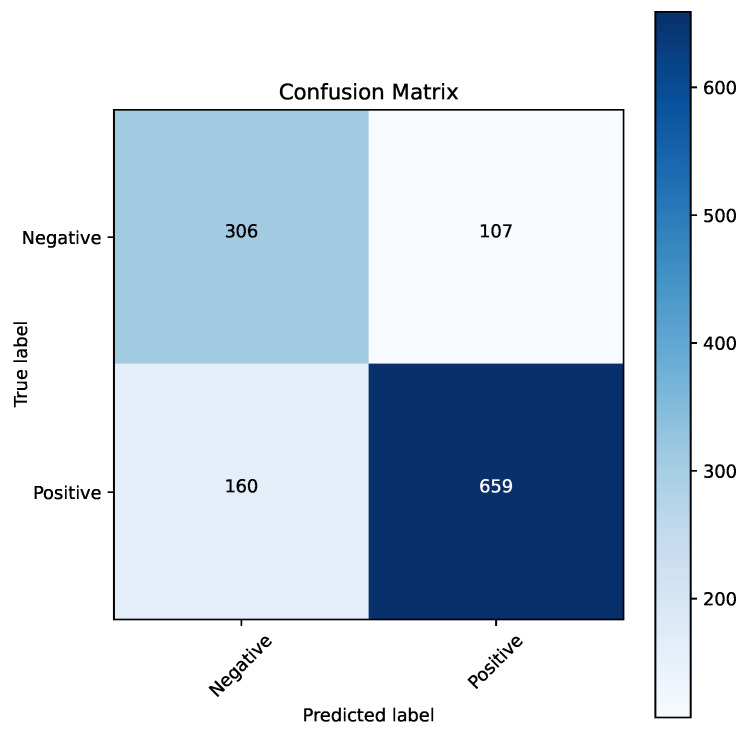
Confusion matrix for XGBoost algorithm for parameter optimization and feature selection using naked mole-rat algorithm and ADASYN oversampling.

**Figure 13 cancers-16-04128-f013:**
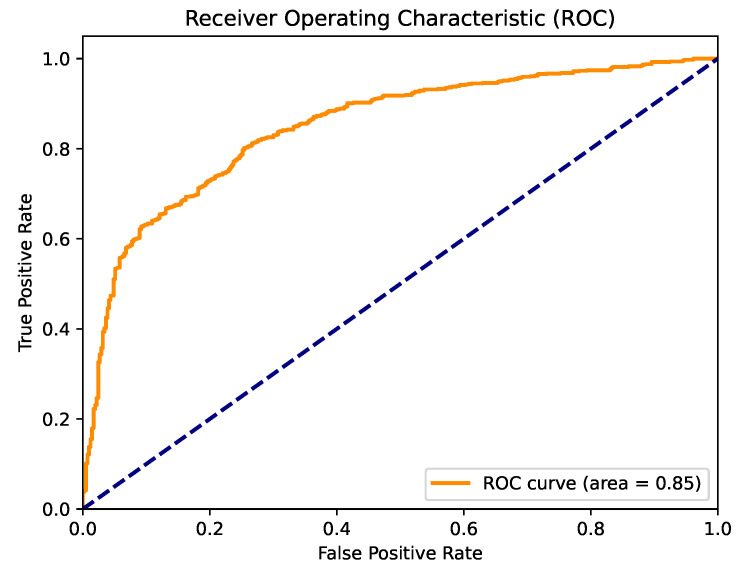
ROC curve for XGBoost algorithm for parameter optimization and feature selection using naked mole-rat algorithm and ADASYN oversampling.

**Figure 14 cancers-16-04128-f014:**
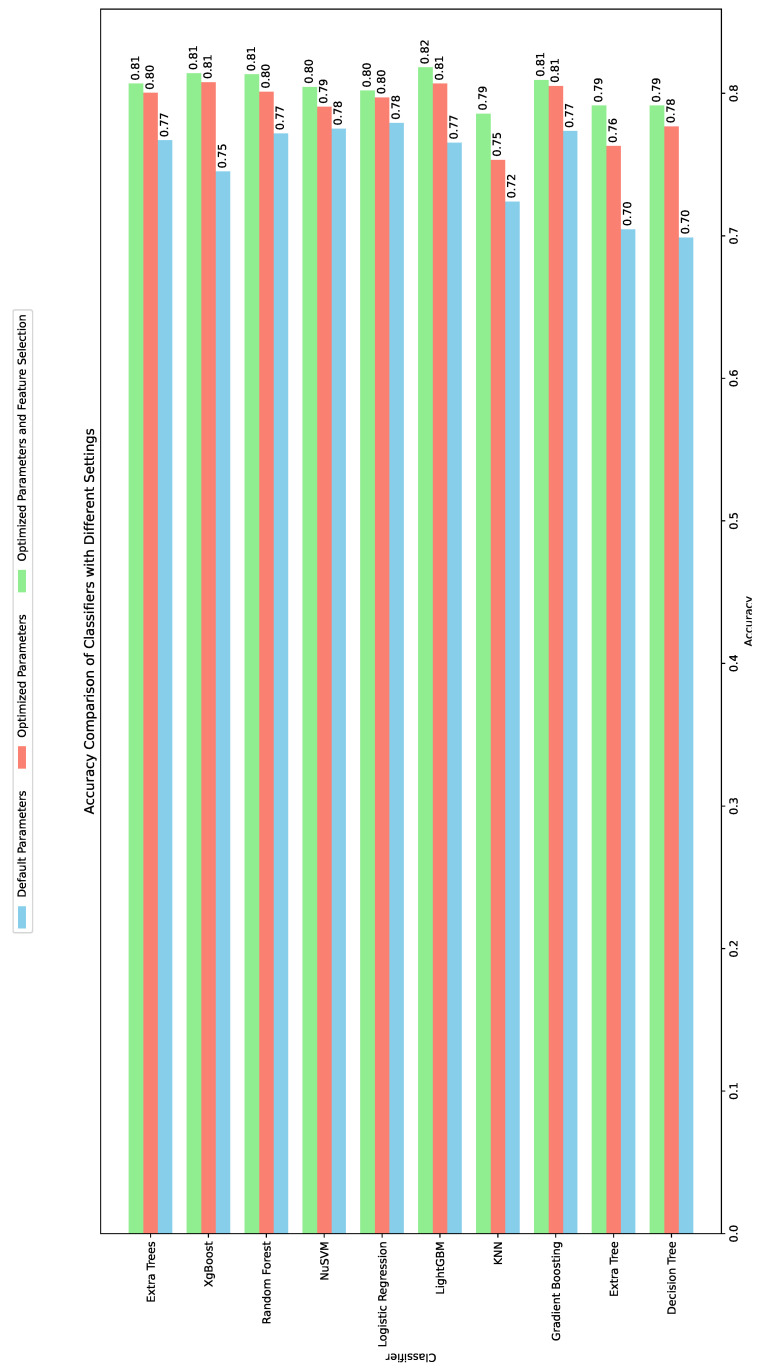
Comparison of accuracy results for individual classifiers in three experiments.

**Table 1 cancers-16-04128-t001:** Thyroid cancer incidence and mortality statistics for 2022 [5].

Population	Number of Cases	Number of Deaths
Oceania	5046	338
Africa	19,740	4936
Latin America and the Caribbean	63,530	4604
Northern America	57,747	2595
Europe	78,552	5902
Asia	596,599	29,132
Total	821,214	47,507

**Table 2 cancers-16-04128-t002:** Overview of studies applying machine learning to thyroid cancer diagnostics.

Article	Database	Model	Accuracy [%]
[15]	Tabular data—25,729 records	Random Forest	90.6
[16]	300k ultrasound images	CNN	89
[17]	MRI from 49 patients	CNN	87
[18]	821 ultrasound images	Mask R-CNN	84
[19]	347 ultrasound images	PCA + FOX optimizer + Random Forest	99.13
[20]	Tabular data—383 records	SVM	99.33
[21]	917 ultrasound images 2352 CT scans	Xception	98.9 97.5
[22]	11,715 histopathological images	Inception-ResNet-v2 VGG-19	94.42 97.34
[23]	14,194 ultrasound images	CNN	57.7
[24]	Tabular data—1232 records	Random Forest	78.01

**Table 3 cancers-16-04128-t003:** Thyroid cancer dataset [25].

Feature	Values	Mean	Median	Std	Mode
Age	The age of the patient	46.6104	47.000	12.4468	55.000
Gender	Patient gender, 0: male, 1: female	0.8377	1	0.3689	1.000
FT3	Triiodothyronine test result	4.4715	4.350	1.1467	4.000
FT4	Thyroxine test result	15.1040	14.510	3.3390	14.000
TSH	Thyroid-stimulating hormone test result	2.0440	1.456	4.6276	0.421
TPO	Thyroid peroxidase antibody test result	72.6771	0.625	203.3970	5.000
TGAb	Thyroglobulin antibodies test result	57.8556	2.690	177.1801	4.000
Site	The nodule location, 0: right, 1: left, 2: isthmus	0.6071	1.000	0.6334	0.000
Echo_pattern	Thyroid echogenicity, 0: even, 1: uneven	0.1088	0.000	0.3115	0.000
Multifocality	If multiple nodules exist in one location, 0: no, 1: yes	0.4610	0.000	0.4987	0.000
Size	The nodule size in cm	1.7313	1.300	1.3140	1.000
Shape	The nodule shape, 0: regular, 1: irregular	0.2070	0.000	0.4053	0.000
Margin	The clarity of nodule margin, 0: clear; 1: unclear	0.6705	1.000	0.4702	1.000
Calcification	The nodule calcification, 0: absent, 1: present	0.3994	0.000	0.4900	0.000
Echo_strength	the nodule echogenicity, 0: none, 1: isoechoic, 2: medium-echogenic, 3: hyperechogenic, 4: hypoechogenic	3.6721	4.000	0.8268	4.000
Blood_flow	The nodule blood flow, 0: normal, 1: enriched	0.3620	0.000	0.4808	0.000
Composition	The nodule composition, 0: cystic, 1: mixed, 2: solid	1.8726	2.000	0.4000	2.000
Multilateral	If nodules occur in more than one location, 0: no, 1: yes	0.7679	1.000	0.4224	1.000
Mal	The nodule malignancy, 0: benign, 1: malignant	0.6648	1.000	0.4723	1.000

**Table 4 cancers-16-04128-t004:** Parameters of selected classifiers.

Classifier	Parameters
Decision tree	criterion, splitter, max_depth, min_samples_split, min_samples_leaf, min_weight_fraction_leaf, max_leaf_nodes, min_impurity_decrease, ccp_alpha
Extra tree	criterion, splitter, max_depth, min_samples_split, min_samples_leaf, min_weight_fraction_leaf, max_leaf_nodes, min_impurity_decrease, ccp_alpha
Gradient boosting	n_estimators, max_depth, loss, learning_rate, criterion, subsample, min_samples_split, min_samples_leaf, min_weight_fraction_leaf, min_impurity_decrease, max_leaf_nodes, ccp_alpha
Knn	metric, weights, n_neighbors
LightGBM	n_estimators, max_depth, min_child_weight, learning_rate, gamma, subsample, colsample_bytree, reg_alpha, reg_lambda, eval_metric, num_leaves, min_child_samples, bagging_freq, feature_fraction, bagging_fraction
Logistic regression	C, max_iter, penalty, dual, fit_intercept, solver, l1_ratio
NuSVM	kernel, nu, degree, gamma
Random forest	criterion, n_estimators, max_depth, min_samples_split, min_samples_leaf, min_weight_fraction_leaf, max_leaf_nodes, min_impurity_decrease, ccp_alpha, bootstrap, max_samples
XgBoost	n_estimators, max_depth, min_child_weight, learning_rate, gamma, subsample, colsample_bytree, reg_alpha, reg_lambda
Extra trees	criterion, n_estimators, max_depth, min_samples_split, min_samples_leaf, min_weight_fraction_leaf, max_leaf_nodes, min_impurity_decrease, ccp_alpha, bootstrap, max_samples

**Table 5 cancers-16-04128-t005:** Results for classifiers with default parameters.

Classifier	Accuracy	F1-Score	AUC	Recall	Precision
Decision Tree	0.6988	0.772	0.66	0.768	0.7767
Extra Tree	0.7045	0.7767	0.67	0.7741	0.7804
Gradient Boosting	0.7735	0.8331	0.83	0.8498	0.8177
KNN	0.724	0.7935	0.77	0.7997	0.7891
LightGBM	0.7654	0.8273	0.82	0.8462	0.8096
Logistic Regression	0.7792	0.8354	0.84	0.8413	0.8306
NuSVM	0.7752	0.8313	0.83	0.8351	0.8285
Random Forest	0.7719	0.8314	0.83	0.8462	0.8175
XgBoost	0.7451	0.8115	0.81	0.8266	0.7975
Extra Trees	0.767	0.8265	0.82	0.8352	0.8186

**Table 6 cancers-16-04128-t006:** Results for classifiers using naked mole-rat algorithm for parameter optimization.

Classifier	Accuracy	F1-Score	AUC	Recall	Precision
Decision Tree	0.7768	0.8258	0.82	0.7973	0.8583
Extra Tree	0.763	0.8108	0.78	0.7692	0.863
Gradient Boosting	0.8052	0.8562	0.85	0.873	0.8404
KNN	0.7533	0.8139	0.80	0.812	0.8169
LightGBM	0.8068	0.8572	0.85	0.873	0.8426
Logistic Regression	0.7971	0.8512	0.84	0.873	0.832
NuSVM	0.7905	0.8418	0.83	0.84	0.8452
Random Forest	0.8011	0.8573	0.85	0.8987	0.8201
XgBoost	0.8077	0.8595	0.85	0.8865	0.8351
Extra Trees	0.8003	0.8571	0.85	0.9011	0.8186

**Table 7 cancers-16-04128-t007:** Results of classifier parameter optimization and feature selection using naked mole-rat algorithm.

Classifier	Number of Features	Accuracy	F1-Score	AUC	Recall	Precision
Decision Tree	10	0.7914	0.8365	0.83	0.8034	0.8735
Extra Tree	14	0.7914	0.8407	0.82	0.8279	0.8557
Gradient Boosting	12	0.8093	0.8588	0.85	0.8731	0.8456
KNN	10	0.7857	0.8407	0.83	0.8522	0.8304
LightGBM	13	0.8182	0.8662	0.86	0.884	0.8497
Logistic Regression	14	0.8019	0.8507	0.83	0.8498	0.8538
NuSVM	13	0.8044	0.8505	0.84	0.8388	0.864
Random Forest	13	0.8133	0.8646	0.85	0.8962	0.8355
XgBoost	11	0.8141	0.8636	0.85	0.884	0.8446
Extra Trees	13	0.8068	0.8571	0.84	0.8718	0.8433

**Table 8 cancers-16-04128-t008:** Comparison of model performance using SMOTE and ADASYN.

Classifier	Oversampling	Accuracy	F1-Score	AUC	Recall	Precision
Decision Tree	SMOTE	0.7646	0.8105	0.82	0.7594	0.8705
	ADASYN	0.7598	0.8052	0.82	0.7484	0.8737
Extra Tree	SMOTE	0.7443	0.7902	0.79	0.7276	0.8683
	ADASYN	0.7614	0.8102	0.80	0.768	0.859
Gradient Boosting	SMOTE	0.7744	0.8232	0.85	0.7913	0.8589
	ADASYN	0.776	0.8221	0.84	0.7814	0.8687
KNN	SMOTE	0.7662	0.8124	0.83	0.7619	0.8707
	ADASYN	0.741	0.7853	0.82	0.7155	0.8729
LightGBM	SMOTE	0.7768	0.8267	0.84	0.801	0.8556
	ADASYN	0.7776	0.8262	0.84	0.7961	0.86
Logistic Regression	SMOTE	0.7566	0.8023	0.83	0.7448	0.8703
	ADASYN	0.7387	0.7831	0.82	0.7118	0.8716
NuSVM	SMOTE	0.7711	0.8151	0.84	0.7607	0.8795
	ADASYN	0.7435	0.7903	0.80	0.7313	0.8634
Random Forest	SMOTE	0.7768	0.8212	0.85	0.7728	0.8773
	ADASYN	0.776	0.819	0.84	0.7643	0.8836
XgBoost	SMOTE	0.7825	0.8307	0.85	0.8034	0.8608
	ADASYN	0.7833	0.8313	0.84	0.8046	0.8609
Extra Trees	SMOTE	0.7679	0.8141	0.83	0.7668	0.8693
	ADASYN	0.759	0.803	0.84	0.7435	0.8762

**Table 9 cancers-16-04128-t009:** Comparison of results with existing literature.

Classifier	Metric	Xi et al. [24]	Current Study
XgBoost	Accuracy	0.7722	0.8141
Random Forest	Accuracy	0.7801	0.8133
LightGBM	Accuracy	-	0.8182

## Data Availability

The data used in this research are publicly available and can be accessed at https://zenodo.org/records/6465436 (accessed on 27 October 2024).

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
