# Peer review of "Explainable Thyroid Cancer Diagnosis Through Two-Level Machine Learning Optimization with an Improved Naked Mole-Rat Algorithm"

_cancers, 2024, doi:10.3390/cancers16244128_

Round 1
Reviewer 1 Report
Comments and Suggestions for Authors
The paper used ten distinct classifiers, including XGBoost, LightGBM, and Random Forest, combined with the naked mole-rat algorithm for parameter optimization and feature selection. The paper contains a thorough evaluation based on simulation.
The research focuses on creating an efficient classification model to assist in thyroid cancer diagnosis. This involves parameter optimization and feature selection across ten classification algorithms. The study introduces the naked mole-rat algorithm for optimization and compares its performance with default classifier settings. Additionally, model explainability is enhanced using the SHAP Values method. Although the paper is technically sound, it needs to be improved to be ready for publication.
Some Suggestions:
1. In the abstract, you do not need to mention the words :Background, Methods, Results, and Conclusions. Write your abstract without mentioning these words for better abstract.
2. The authors should consider more recent papers in the introduction section when discussing the rise of thyroid cancer and its rapid growth. It could be beneficial to see what the percentage is in 2024, as the cited numbers in the paper are from 2016-2017 in reference [2] and [3].
3. Figure 1 is visually appealing but it is difficult to follow and understand and it is from 2022.
4. It is better to have introduction and literature review (or related works) in separate sections to make it easy for the reader. Also, It would be beneficial to have a summary table for related works. It would be beneficial to include a problem statement and motivational scenario.
5. The authors did not mention the use of deep learning in the field of Thyroid Cancer Diagnosis. It is better to include some papers that use this advanced technology.
6. The dataset contains 1,232 records, and it is kind of small number of records to train an ML model and it could cause overfitting. How the authors handle this?
7. It is good that Table 3 lists the parameters of each ML models but what about their values?
8. Wha are the differences between Table 5 and Table 6. It is not clear.
9. The classification accuracy of the proposed model is 81.82%. However, there are researchers who have outperformed this research result, achieving more than 90% in the field of Thyroid Cancer. What are the reasons for such low accuracy compared with other studies?
Author Response
Thank you for taking the time to review our manuscript. We greatly appreciate your valuable
feedback and suggestions, which have helped improve the quality of our work.
Comments 1:
In the abstract, you do not need to mention the words :Background, Methods, Results, and
Conclusions. Write your abstract without mentioning these words for better abstract.
Response 1:
Thank you for your comment. The author of the article followed the default MDPI template in
Overleaf, which includes the sections: Background, Methods, Results, and Conclusions.
Below is the revised version of the abstract:
Modern technologies, particularly artificial intelligence methods such as machine learning,
hold immense potential in supporting doctors with cancer diagnostics. This research focuses
on developing new machine learning models designed to assess the malignancy of thyroid
tumors. The study utilized a novel dataset released in 2022, containing data collected at
Shengjing Hospital of China Medical University. The dataset comprises 1,232 records
described by 19 features. In this research, 10 well-known classifiers, including XGBoost,
LightGBM, and Random Forest, were employed to evaluate the malignancy of thyroid
tumors. A key innovation of this study is the application of the Naked Mole-Rat Algorithm for
parameter optimization and feature selection within the individual classifiers. Among the
models tested, the LightGBM classifier demonstrated the highest performance, achieving a
classification accuracy of 81.82% and an F1-score of 86.62%, following two-level parameter
optimization and feature selection using the Naked Mole-Rat Algorithm. Additionally,
explainability analysis of the LightGBM model was conducted using SHAP values, providing
insights into the decision-making process of the model.
Comments 2:
The authors should consider more recent papers in the introduction section when discussing the rise
of thyroid cancer and its rapid growth. It could be beneficial to see what the percentage is in 2024, as
the cited numbers in the paper are from 2016-2017 in reference [2] and [3].
Response 2:
Thank you for your comment. You are correct that the article initially lacked references to the latest
statistics on thyroid cancer incidence and mortality. While comprehensive statistics for 2023 and
2024 are not yet fully available, some data from the USA were identified and incorporated into the
article. The author retained information from previous years while adding new data to allow readers
to form their own opinions. Additionally, the author considers the data provided by GLOBOCAN,
despite being from 2022, to be highly reliable. This information is presented in Table 1.
The following addition was made to the body of the article (lines 43-46):
According to the latest research [6], 43,720 cases of thyroid cancer were diagnosed in the USA in
2023, representing a 313% increase in incidence over recent decades. The American Cancer Society
predicts a slight increase in 2024, with the number of detected cases expected to reach 44,020 [7].
References:
6. Boucai, L.; Zafereo, M.; Cabanillas, M.E. Thyroid Cancer: A Review. JAMA 2024, 331, 425.
https://doi.org/10.1001/jama.2023.26348.
7. Facts about Thyroid Cancer (accessed on 27 October 2024). https://www.thyca.org/about/thyroidcancer-
facts/ .
Comments 3:
Figure 1 is visually appealing but it is difficult to follow and understand and it is from 2022.
Response 3:
Thank you for highlighting this point. While this figure is indeed challenging to analyze, it is
highly valuable as it encompasses global data. To enhance accessibility, I have moved it to the
supplementary materials, ensuring that readers can easily reference it without needing to
search for these statistics. Despite being from 2022, the GLOBOCAN data is, in the author’s
opinion, one of the most reliable sources for global cancer incidence and mortality rates.
The following addition was made to the main text of the article (lines 38-41):
Data from 2022 [5] show that thyroid cancer ranks as the third most common cancer in
China and Saudi Arabia, fourth in Mexico, and fifth in Brazil and Turkey. Detailed incidence
data by country are available in the supplementary materials.
References:
5. Ferlay, J.; Ervik, M.; Lam, F.; Colombet, M.and Mery, L.; Piñeros, M.and Znaor, A.;
Soerjomataram, I.; Bray, F. Global Cancer 546 Observatory: Cancer Today (version 1.1).
Lyon, France: International Agency for Research on Cancer (accessed on 27 October 547
2024). https://gco.iarc.who.int/today .
Comments 4:
It is better to have introduction and literature review (or related works) in separate sections to make
it easy for the reader. Also, It would be beneficial to have a summary table for related works. It
would be beneficial to include a problem statement and motivational scenario.
Response 4:
Thank you for pointing this out. Separating the introduction and literature review has indeed
enhanced the article's readability. Accordingly, a new section titled Literature Review has been
introduced. Additionally, as suggested, a table summarizing the studies discussed in the literature
review has been added—Table 2: Overview of Studies Applying Machine Learning to Thyroid Cancer
Diagnostics.
On lines 115-123, text summarizing this table has also been added:
A summary of selected literature on the application of machine learning methods to thyroid cancerrelated
issues is presented in Table 2. It is important to note that these studies were conducted using
various types of data (e.g., tabular data, ultrasound, MRI, CT, histopathological images) and pursued
different objectives, such as disease diagnosis, prediction of metastases, assessment of recurrence
risk, and evaluation of tumor malignancy. Consequently, direct comparison of the results is not
feasible. However, it is worth highlighting that the number of studies in this area remains relatively
small. Further research is essential to develop effective machine learning models for diagnosing and
supporting the treatment of patients with suspected or confirmed thyroid cancer.
In lines 129-142, an explanation was added regarding the statement of the problem and the
motivation for conducting this research:
The primary motivation for this research was the necessity of developing new machine learning
models capable of effectively assessing the malignancy of thyroid tumors. The study utilized the
dataset from [24] to ensure that future researchers can compare their work, facilitating the
development of more effective diagnostic models. It is worth emphasizing that the number of
scientific studies focusing on this issue is relatively small, underscoring the need for continued efforts
in this field to design models that could eventually be applied in clinical practice. Furthermore,
previous studies have often overlooked the aspect of model explainability, which is essential for the
application of machine learning in medical diagnostics. An additional motivation for the author was
to advance modern bio-inspired metaheuristic methods, such as the Naked Mole-Rat Algorithm, to
demonstrate their effectiveness and highlight their value in addressing machine learning challenges in
medical diagnostics. This research aims to contribute to progress in both medical diagnostics and bioinspired metaheuristics by addressing advanced optimization problems, such as parameter tuning for
machine learning models and feature selection.
Comments 5:
The authors did not mention the use of deep learning in the field of Thyroid Cancer Diagnosis. It is
better to include some papers that use this advanced technology.
Response 5:
Thanks for highlighting this fact. Added three additional articles focusing exclusively on deep learning
methods in thyroid cancer diagnosis. Added the following text on lines 90-109:
The study by [21] employed multi-channel convolutional networks, specifically the Xception network,
for diagnosing thyroid cancer. The research was conducted using both ultrasound and computed
tomography (CT) images. The dataset included 917 ultra- sound images and 2,352 CT images. The
classification accuracy achieved was 98.9% for ultrasound images and 97.5% for CT images. In works
[22], the authors conducted their research on a dataset comprising histopathological images, which
included 11,715 images from 806 patients. For classification, they utilized deep learning methods,
specifically the Inception-ResNet-v2 and VGG-19 convolutional networks. The study focused on a
multiclass classification task, addressing the following classes: normal tissue, adenoma, nodular
goiter, papillary thyroid carcinoma (PTC), follicular thyroid carcinoma (FTC), medullary thyroid
carcinoma (MTC), and anaplastic thyroid carcinoma (ATC). The VGG-19 model achieved a
classification accuracy of 97.34%, while Inception-ResNet-v2 achieved 94.42%. These results
demonstrate that deep learning models can be effectively applied to complex diagnostic challenges in thyroid cancer.The researchers in [23] conducted an intriguing study aimed at predicting the
BRAFV600E mutation in thyroid cancer using ultrasound images. The study utilized a dataset
comprising 14194 ultrasound images and employed several pre-trained deep learning models,
including AlexNet, GoogLeNet, SqueezeNet, and Inception-ResNet-v2. The best-performing model
achieved a classification accuracy of 57.7% and an AUC of 64.6%.
References:
21. Zhang, X.; Lee, V.C.S.; Rong, J.; Liu, F.; Kong, H. Multi-channel convolutional neural network
architectures for thyroid cancer detection. PLOS ONE 2022, 17, e0262128.
https://doi.org/10.1371/journal.pone.0262128. 587
22. Wang, Y.; Guan, Q.; Lao, I.; Wang, L.; Wu, Y.; Li, D.; Ji, Q.; Wang, Y.; Zhu, Y.; Lu, H.; et al. Using
deep convolutional neural networks for multi-classification of thyroid tumor by histopathology: a
large-scale pilot study. Annals of Translational Medicine 2019, 7, 468–468.
https://doi.org/10.21037/atm.2019.08.54. 590
23. Yoon, J.; Lee, E.; Koo, J.S.; Yoon, J.H.; Nam, K.H.; Lee, J.; Jo, Y.S.; Moon, H.J.; Park, V.Y.; Kwak, J.Y.
Artificial intelligence to predict the BRAFV600E mutation in patients with thyroid cancer. PLOS ONE
2020, 15, e0242806. https://doi.org/10.1371/journal.pone. 5920242806.
I also emphasized that deep learning methods were not applied in these studies due to the small
sample size and limited number of features in the dataset.
The following was added in lines 170-171:
Deep learning methods were not used in these studies due to the small sample size and the limited
number of features in the dataset.
Comments 6:
The dataset contains 1,232 records, and it is kind of small number of records to train an ML model
and it could cause overfitting. How the authors handle this?
Response 6:
Thank you for pointing this out. While the dataset is relatively small, and there is a risk of overfitting,
it is worth noting that it is not extremely limited in size. For instance, in the case of omics data, the
number of samples often ranges between 10 and 100 [1]. To mitigate the risk of overfitting, the
author, following the approach used by the creators of the dataset [2], employed 10-fold crossvalidation.
While this is not a perfect solution, the study’s limitations clearly highlight the necessity of
obtaining larger datasets for future research.
References:
1) Kirpich A, Ainsworth EA, Wedow JM, Newman JRB, Michailidis G, McIntyre LM (2018)
Variable selection in omics data: A practical evaluation of small sample sizes. PLoS ONE
13(6): e0197910. https://doi.org/10.1371/journal.pone.0197910
2) Xi, N.M.; Wang, L.; Yang, C. Improving the diagnosis of thyroid cancer by machine learning
and clinical data. Scientific Reports
2022, 12. https://doi.org/10.1038/s41598-022-15342-z.
The following text was added in lines 494 to 502:
The research was conducted using a single dataset from China, comprising 1,232 data samples. This
dataset may be insufficient to ensure the model's high generalization ability and could lead to a
degree of overfitting. The next step in developing these models is to create a significantly larger
dataset. Furthermore, acquiring data from diverse geographical regions would be highly beneficial, as
it would allow consideration of local factors such as diet, environmental conditions, and access to
healthcare. Unfortunately, such data is not currently publicly available. Nevertheless, the author
remains optimistic that more datasets will become accessible in the future, enabling further
advancements in this research.
Additionally, the following was added on lines 258-261:
The use of 10-fold cross-validation not only facilitated the comparison of the obtained results with
those of [24] but also helped reduce overfitting in the [46] models. However, the author highlights in
the study's limitations that further development of these models will require access to larger
datasets.
References:
24. Xi, N.M.; Wang, L.; Yang, C. Improving the diagnosis of thyroid cancer by machine learning and
clinical data. Scientific Reports 2022, 12. https://doi.org/10.1038/s41598-022-15342-z.
46. Kaliappan, J.; Bagepalli, A.R.; Almal, S.; Mishra, R.; Hu, Y.C.; Srinivasan, K. Impact of Cross-
Validation on Machine Learning Models for Early Detection of Intrauterine Fetal Demise. Diagnostics
2023, 13, 1692. https://doi.org/10.3390/diagnostics13101692.
Comments 7:
It is good that Table 3 lists the parameters of each ML models but what about their values?
Response 7:
Thank you for this comment. You are correct that the table does not include value ranges for the
individual models. To enhance the readability and clarity of the article, this information has been
provided in the supplementary materials.
The following text has been added to lines 174-176:
The supplementary materials also include the ranges of values for the individual parameters used in
the optimization process.
The following has been added in the supplementary materials:
6. Optimized parameters of individual classifiers
6.1. Decision tree
6.2. Extra Tree
Comments 8:
What are the differences between Table 5 and Table 6. It is not clear.
Response 8:
Thank you for pointing this out. Table 5 shows the results obtained using the default classifier
parameter settings provided by libraries such as scikit-learn, XGBoost, and LightGBM. In contrast,
Table 6 presents the results after optimizing the classifier parameters using the Naked Mole-Rat
Algorithm. Additional explanations regarding these aspects have been included in the article text.
The following text has been added to lines 300-305:
In the first experiment, the author chose not to apply parameter optimization in order to observe the
performance of the classifiers with their default settings, as configured in libraries such as scikit-learn,
XGBoost, and LightGBM.. In subsequent sections of the results, the improvements in model
performance achieved through parameter optimization using the Naked Mole-Rat Algorithm will be
highlighted.
The following text has been added to lines 318-320:
In contrast to the experiments in the previous section, these experiments involved optimizing the
parameters of the machine learning models to enhance their classification performance.
Comments 9:
The classification accuracy of the proposed model is 81.82%. However, there are researchers who
have outperformed this research result, achieving more than 90% in the field of Thyroid Cancer.
What are the reasons for such low accuracy compared with other studies?
Response 9:
Thank you for this comment. It is important to emphasize that comparing individual studies based
solely on the classification accuracy of their models (essentially a single metric) is not feasible. This is
particularly true when studies utilize different types of data—such as tabular data, ultrasound, MRI,
CT scans, or histopathological images—especially if these datasets vary in size and are often not
publicly available. Consequently, it is impossible to directly compare newly designed models with
those from previous literature by running experiments on the same datasets. For this reason, the
author focused solely on comparisons with the study conducted by Xi et al. in 2022. This work
provided publicly available experimental data, which allowed for the replication of experiments using
the same approach to train-test splitting (10-fold cross-validation). This methodology enables a
meaningful comparison of the obtained results and provides future researchers with a framework for
evaluating their models against the results presented in this study.
The following text has been added to lines 477-483:
In this discussion, we avoided comparing our results with the studies listed in Table 9, as those studies
were conducted on datasets with entirely different structures (e.g., ultrasound, MRI, CT scans,
histopathological images). Models that perform well on tabular data may not necessarily achieve
similar performance on image-based data. Additionally, most of the datasets referenced in these
publications are not publicly available, making it impossible to evaluate newly designed models
against them.

Reviewer 2 Report
Comments and Suggestions for Authors
This paper uses the 10 existing classification models from machine learning techniques for diagnosing this thyroid cancer by assessing tumor malignancy, and the 10 existing methods have decision tree extra tree, gradient boosting, nearest neighbor algorithm, LightGBM, logistic regression, NU-SVM support vector machine, random forest, XGBoost, and extra trees. The experimental data is from Shengjing Hospital of China Medical University, consisting of 1,232 records. Their performance indicators have classification accuracy, F1-score, confusion matrices, and ROC curves with using SHAP values. For the experimental results, it is showed that they achieved a classification accuracy of 81.82% and an F1-score of 86.62%. It is concluded that machine learning algorithms can have an advanced bioinspired metaheuristic, demonstrate effectiveness in assessing tumor malignancy in thyroid cancer patients. After my carefully reviewing, although this paper has an interesting topic in medical industry, it has some problems that the problematic issues should be firstly addressed and repaired in the revised manuscript, as follows:
1. Initially, after carefully reviewing the manuscript, although the paper is interesting and deserves consideration but I'm not sure about the novelty and contributions in the methodology originality of this paper because related studies can be found in other researches. And I do not see a clear discussion on the important point "their methodology related to earlier works" - do we have anything new or not? Or perhaps, could they tell us what material is new? And the new material is important? Thus, please more clearly tell us about the above problems when compared to past researches. Thus, this paper should firstly point out its specific contribution and strength when compared to past studies, because I think that this paper has not well enough for specific technical contribution, particularly for technique or method innovation; and it is necessary that the author must be improved with much more specificity and novelty of what is done.
2. Also, significance of the research and motivation for this study is not clearly stated.
3. It seems that there is no traditional section focusing on literature reviews. By presuming that the methods and results represent the literature review, the authors may create some confusion to general readers. So, I suggest that there is a need for adding a section for a thorough literature review.
4. For Section 3: Results, I suggest the authors should add a table to do the descriptive statistics for the used data of all attributes, including conditional attributes and decisional attribute for the cause-effect relationship. That is very useful and helpful for improving the readability of the paper.
5. Thus, please also clearly state all the attributes used in the dataset, including what is the class classification, how much instances for this decision class? Maybe, it is possible that it has a class imbalanced problem. Now it is unclear.
6. The paper seems to have some theoretical. Thus, it is recommended that more potential application results or discussions for practical applications on the empirical results would enhance the paper. Also, please make a more explanation for the main results of application views into the Conclusions parts for valuing the study.
7. Please check all literature, and they should be completely followed with the journal style and format of cancers journal.
8. The language in the paper is needed for further proofreading.
Author Response
Thank you for taking the time to review our manuscript. We greatly appreciate your valuable
feedback and suggestions, which have helped improve the quality of our work.
Comments 1:
Initially, after carefully reviewing the manuscript, although the paper is interesting and deserves
consideration but I'm not sure about the novelty and contributions in the methodology originality of
this paper because related studies can be found in other researches. And I do not see a clear
discussion on the important point "their methodology related to earlier works" - do we have
anything new or not? Or perhaps, could they tell us what material is new? And the new material is
important? Thus, please more clearly tell us about the above problems when compared to past
researches. Thus, this paper should firstly point out its specific contribution and strength when
compared to past studies, because I think that this paper has not well enough for specific technical
contribution, particularly for technique or method innovation; and it is necessary that the author
must be improved with much more specificity and novelty of what is done.
Response 1:
Thank you for your comment. Indeed, the novelty of the research has not been sufficiently
emphasized.
The following text has been added to the discussion in lines 418–450:
The primary novelty of this research is the use of the Naked Mole-Rat Algorithm, a relatively new
population-based algorithm within the bio-inspired metaheuristics category. To date, this algorithm
has been predominantly applied to standard benchmark problems, such as the optimization of
multidimensional functions in the CEC conference. Notably, bio-inspired metaheuristics have been
rarely utilized in thyroid cancer detection problems—only one study using the Fox Optimizer
algorithm was found in the literature. This highlights the need for further research in this field,
especially since these methods have demonstrated high effectiveness in numerous medical
classification tasks. In this study, the Naked Mole-Rat Algorithm is applied for the first time to
optimize machine learning model parameters and to perform feature selection. In the first
optimization task—classifier parameter selection—a method was developed to encode parameters as
real numbers, such that each individual in the population represented a classifier capable of
diagnosing thyroid cancer. For certain parameters, such as the metric in the nearest neighbor
algorithm (Euclidean, Manhattan, Chebyshev), text-based values were encoded as real numbers to
align with the requirements of the Naked Mole-Rat Algorithm. This approach resulted in a one-level
optimization model. The second part of the research introduces a novel two-level optimization
approach, which simultaneously optimizes both parameters and feature selection. In this case, each
individual was extended with an additional feature vector, where each feature had a value ranging
from 0 to 1. Values between 0 and 0.5 indicated the rejection of a feature, while values between 0.5
and 1 signified feature selection for the model. This two-level approach allowed for the exploration of
appropriate classifiers within the multidimensional space of parameters and features, presenting a
complex optimization problem. It is important to note that this represents a significant advancement
compared to the study [24], where parameter selection and feature selection were not discussed.
Additionally, this research involved the use of 10 distinct classification models, ensuring a
comprehensive study that included both well-established algorithms based on metrics and rules, as
well as effective gradient boosting models and support vector machines for small and medium-sized
datasets. Furthermore, the interpretability of the models was emphasized through the use of SHAP
values, a critical aspect for implementing algorithms in clinical practice. Additional experiments were
conducted with oversampling techniques, including SMOTE and ADASYN, to assess the potential of
these methods for producing highly efficient classifiers. The description above highlights the high
novelty of this research and the thoroughness of the experiments conducted.
Comments 2:
Also, significance of the research and motivation for this study is not clearly stated.
Response 2:
Thank you for this comment. The significance and motivation behind this research have been
emphasized.
The following text has been added to lines 129–142:
The primary motivation for this research was the necessity of developing new machine learning
models capable of effectively assessing the malignancy of thyroid tumors. The study utilized the
dataset from [24] to ensure that future researchers can compare their work, facilitating the
development of more effective diagnostic models. It is worth emphasizing that the number of
scientific studies focusing on this issue is relatively small, underscoring the need for continued efforts
in this field to design models that could eventually be applied in clinical practice. Furthermore,
previous studies have often overlooked the aspect of model explainability, which is essential for the
application of machine learning in medical diagnostics. An additional motivation for the author was
to advance modern bio-inspired metaheuristic methods, such as the Naked Mole-Rat Algorithm, to
demonstrate their effectiveness and highlight their value in addressing machine learning challenges in
medical diagnostics. This research aims to contribute to progress in both medical diagnostics and bioinspired metaheuristics by addressing advanced optimization problems, such as parameter tuning for
machine learning models and feature selection.
Comments 3:
It seems that there is no traditional section focusing on literature reviews. By presuming that the
methods and results represent the literature review, the authors may create some confusion to
general readers. So, I suggest that there is a need for adding a section for a thorough literature
review.
Response 3:
Thank you for this comment. A separate section titled "Literature Review" has been created.
Additionally, three more studies related to deep learning and thyroid cancer have been added in
lines 90–109.
The study by [21] employed multi-channel convolutional networks, specifically the Xception network,
for diagnosing thyroid cancer. The research was conducted using both ultrasound and computed tomography (CT) images. The dataset included 917 ultra- sound images and 2,352 CT images. The
classification accuracy achieved was 98.9% for ultrasound images and 97.5% for CT images. In works
[22], the authors conducted their research on a dataset comprising histopathological images, which
included 11,715 images from 806 patients. For classification, they utilized deep learning methods,
specifically the Inception-ResNet-v2 and VGG-19 convolutional networks. The study focused on a
multiclass classification task, addressing the following classes: normal tissue, adenoma, nodular
goiter, papillary thyroid carcinoma (PTC), follicular thyroid carcinoma (FTC), medullary thyroid
carcinoma (MTC), and anaplastic thyroid carcinoma (ATC). The VGG-19 model achieved a
classification accuracy of 97.34%, while Inception-ResNet-v2 achieved 94.42%. These results
demonstrate that deep learning models can be effectively applied to complex diagnostic challenges in
thyroid cancer.The researchers in [23] conducted an intriguing study aimed at predicting the
BRAFV600E mutation in thyroid cancer using ultrasound images. The study utilized a dataset
comprising 14194 ultrasound images and employed several pre-trained deep learning models,
including AlexNet, GoogLeNet, SqueezeNet, and Inception-ResNet-v2. The best-performing model
achieved a classification accuracy of 57.7% and an AUC of 64.6%.
References:
21. Zhang, X.; Lee, V.C.S.; Rong, J.; Liu, F.; Kong, H. Multi-channel convolutional neural network
architectures for thyroid cancer detection. PLOS ONE 2022, 17, e0262128.
https://doi.org/10.1371/journal.pone.0262128. 587
22. Wang, Y.; Guan, Q.; Lao, I.; Wang, L.; Wu, Y.; Li, D.; Ji, Q.; Wang, Y.; Zhu, Y.; Lu, H.; et al. Using
deep convolutional neural networks for multi-classification of thyroid tumor by histopathology: a
large-scale pilot study. Annals of Translational Medicine 2019, 7, 468–468.
https://doi.org/10.21037/atm.2019.08.54. 590
23. Yoon, J.; Lee, E.; Koo, J.S.; Yoon, J.H.; Nam, K.H.; Lee, J.; Jo, Y.S.; Moon, H.J.; Park, V.Y.; Kwak, J.Y.
Artificial intelligence to predict the BRAFV600E mutation in patients with thyroid cancer. PLOS ONE
2020, 15, e0242806. https://doi.org/10.1371/journal.pone. 5920242806.
A table summarizing the literature review has also been included.
The following text has been added in lines 115–123:
A summary of selected literature on the application of machine learning methods to thyroid cancerrelated issues is presented in Table 2. It is important to note that these studies were conducted using various types of data (e.g., tabular data, ultrasound, MRI, CT, histopathological images) and pursued different objectives, such as disease diagnosis, prediction of metastases, assessment of recurrence risk, and evaluation of tumor malignancy. Consequently, direct comparison of the results is not feasible. However, it is worth highlighting that the number of studies in this area remains relatively
small. Further research is essential to develop effective machine learning models for diagnosing and
supporting the treatment of patients with suspected or confirmed thyroid cancer.
Comments 4:
For Section 3: Results, I suggest the authors should add a table to do the descriptive statistics for the
used data of all attributes, including conditional attributes and decisional attribute for the causeeffect relationship. That is very useful and helpful for improving the readability of the paper.
Response 4:
Thank you for your comment. Descriptive statistics, including the mode, mean, median, and standard
deviation, have been added to Table 3 to provide a detailed overview of the dataset.
The corrected table looks as follows:
Comments 5:
Thus, please also clearly state all the attributes used in the dataset, including what is the class
classification, how much instances for this decision class? Maybe, it is possible that it has a class
imbalanced problem. Now it is unclear.
Response 5:
Thank you for this comment. This is a crucial observation. Unbalanced medical data is a common
issue that can significantly impact the efficiency of classification models. In this case, the dataset
contains 819 malignant samples and 413 benign samples, demonstrating its imbalance.
However, as shown in the confusion matrices in Figures 5, 7, and 9, the models handle the
classification task effectively. This indicates that the impact of the dataset's imbalance on the
research outcomes is limited.
In many studies, techniques to balance datasets are commonly employed, particularly when models
struggle to learn effectively with imbalanced data. In this research, the models performed well
despite the imbalance; however, to ensure thoroughness, I also applied oversampling techniques.
The two most popular methods, SMOTE and ADASYN, were selected.
The author takes a cautious stance on these methods, as many scientific studies commit a critical
methodological error by performing oversampling before splitting the dataset into training and test
sets. This often results in artificially inflated classification accuracy—sometimes 20–30% higher—
which would quickly be disproven in clinical practice. Any improvements achieved through
oversampling should realistically be within a few percentage points. Moreover, the complex
structure and causal relationships in medical data can result in synthetic samples that are not
representative of real-world cases.
In this research, oversampling techniques were applied correctly—only on the training set.
Unfortunately, this did not result in improved outcomes; in fact, the results declined slightly by a few
percentage points.
A new chapter has been added to the article: "4.5 Application of SMOTE and ADASYN for Data
Balancing".
The following text has also been added in lines 388-408:
In the final part of the experiment, the best models obtained through parameter optimization and
feature selection using the Naked Mole-Rat Algorithm were applied, and the most popular
oversampling techniques—SMOTE and ADASYN—were used. The goal of applying these methods was
to balance the training set, ensuring that the model could effectively classify both benign and
malignant tumors. In Table 8, it can be seen that the oversampling techniques used—SMOTE and
ADASYN—did not result in an improvement in model classification accuracy, and even led to a slight
decrease in effectiveness. In this experiment, the XGBoost algorithm performed the best, achieving a
classification accuracy of 0.7833 and an F1-score of 0.8313. The best results achieved through
parameter optimization and feature selection alone were obtained by the LightGBM algorithm, with a
classification accuracy of 0.8182 and an F1-score of 0.8662. Therefore, it can be concluded that the
use of oversampling methods did not im- prove the performance of the algorithms. This may be due
to the complexity of the medical data or the relatively small dataset, which may limit the full potential
of these algorithms. Figure 12 shows the confusion matrix for the best model from the oversampling
experiment. The ADASYN model performed slightly better than the SMOTE technique, achieving a
final classification accuracy of 0.7833. Additionally, Figure 13 presents the ROC curve for this model,
which demonstrates a high AUC value of 0.84. Confusion matrices and ROC curves for all models listed
in Table 8 can be found in the supplementary materials.
Additionally, the following text has been incorporated into lines 155–157:
The classification problem addressed in this study is binary, distinguishing between malignant and
benign cases. The dataset comprises 819 malignant samples and 413 benign samples.
Comments 6:
The paper seems to have some theoretical. Thus, it is recommended that more potential application
results or discussions for practical applications on the empirical results would enhance the paper.
Also, please make a more explanation for the main results of application views into the Conclusions
parts for valuing the study.
Response 6:
Thank you for your comment. Information regarding the potential further applications of the
naked mole-rat algorithm has been added to the discussion section.
The following passage has been included in lines 511-519:
The research conducted demonstrates the significant potential of the naked mole-rat
algorithm. Machine learning models built using this algorithm showed high classification
accuracy, highlighting its potential for use in various advanced machine learning problems.
These include the optimization of ensemble classifiers built using stacking methods, training
neural networks (as an alternative to the standard backpropagation technique), and
searching for optimal neural network architectures. Additionally, the high performance
observed on the tabular dataset suggests that similar results could be achieved with more
complex data, such as ECG, EEG signals, or medical imaging data like ultrasound, X-ray, or
CT scans.
In Conclusions, the following was added in lines 525-527:
The use of the naked mole-rat algorithm for two-level optimization of machine learning model
parameters and feature selection led to the achievement of high performance results.
Comments 7:
Please check all literature, and they should be completely followed with the journal style and format
of cancers journal.
Response 7:
Thank you for this comment. All references in the manuscript were carefully analyzed.
Comments 8:
The language in the paper is needed for further proofreading.
Response 8:
Thank you for your comment. The article has been thoroughly reviewed, and its linguistic quality has
been enhanced. Changes made to the article text are in red.

Round 2
Reviewer 1 Report
Comments and Suggestions for Authors
We thank the authors for the responses, and they have covered most of my comments, however, I still have two comments as follows:
1- In abstract, what does the author mean by the phrase “new machine learning” does this mean that the author has developed a new ML algorithm? or does it mean that the author has enhanced existing machine learning?
2- The author should consider moving the research objectives or contribution from the literature review to the end of the introduction section and remove it from the literature review.
Author Response
Thank you for taking the time to review our manuscript. We greatly appreciate your valuable
feedback and suggestions, which have helped improve the quality of our work.
Comments 1:
In abstract, what does the author mean by the phrase “new machine learning” does this mean that
the author has developed a new ML algorithm? or does it mean that the author has enhanced
existing machine learning?
Response 1:
Thank you for your comment. I agree that the phrase "new machine learning" is imprecise. The
intended meaning was to indicate the improvement of existing methods using the Naked Mole-Rat
Algorithm. The abstract has been revised accordingly, with changes made in lines 10–12.
This study explores the enhancement of popular machine learning methods using a bio-inspired
algorithm—the Naked Mole-Rat Algorithm (NMRA)
Comments 2:
The author should consider moving the research objectives or contribution from the literature review
to the end of the introduction section and remove it from the literature review.
Response 2:
Thank you very much for your comment. The research objectives have been relocated to the
introduction section and can now be found on lines 54–63:
This research aims to contribute to progress in both medical diagnostics and bioinspired
metaheuristics by addressing advanced optimization problems, such as parameter tuning for
machine learning models and feature selection. The main objectives of this research were:
• Developing an effective classification model to support thyroid cancer diagnosis by optimizing
parameters and selecting features.
• Adapting and applying the naked mole-rat algorithm for parameter optimization and feature
selection in classification models, and comparing its performance to that of default classifier settings.
• Focusing on model explainability by utilizing the SHAP Values method.
